# Advanced radiotherapy technique in hepatocellular carcinoma with portal vein thrombosis: Feasibility and clinical outcomes

**Chonlakiet Khorprasert[1], Kanokphorn Thonglert[2], Petch Alisanant[2], Napapat Amornwichet**[1]*

**1** Faculty of Medicine, Division of Radiation Oncology, Department of Radiology, Chulalongkorn University, Bangkok, Thailand, **2** Division of Radiation Oncology, Department of Radiology, King Chulalongkorn Memorial Hospital, Bangkok, Thailand

* Napapat.a@chula.ac.th, Napapat.amornwichet@gmail.com

**Data Availability Statement:** All relevant data are within the manuscript and its Supporting Information files.

## Abstract

### Background

In Thailand, individuals with hepatocellular carcinoma (HCC) who develop portal vein tumor thrombosis (PVTT) have a restricted treatment option because to the extent of the disease, poor underlying liver function, and non-coverage of immuno/targeted therapy. Radiotherapy (RT) plays an increasingly important function in these patients. To investigate the feasibility, efficacy, and adverse event rates, we performed a retrospective analysis of patients with HCC with PVTT who underwent 3-dimensional conformal radiation (3DCRT), intensity-modulated radiation (IMRT), volumetric-modulated radiotherapy (VMAT), and stereotactic body radiotherapy (SBRT) in a single—institution.

### Objectives

To examine clinical results in terms of overall survival (OS), local control (LC), response of primary tumor and PVTT, hepatic and gastrointestinal adverse reaction, and prognosis variables for OS and LC.

### Materials and methods

Between July 2007 and August 2019, non-metastatic HCC with PVTT patients treated with RT were retrospectively reviewed and evaluated.

### Results

The analysis included data from 160 patients. The mean age of the patients was 60.8 years ((95% CI 58.2–62.0). The median diameter of the tumor was 7.7 cm (range: 1–24.5). 85 (54.5%) individuals had PVTT in the main or first branch. At 1.8–10 Gy per fraction, the mean biologically effective dose (BED) as $\alpha/\beta$ ratio of 10 was 49.6 (95% CI 46.7–52.5) $Gy_{10}$. The median survival time was 8.3 (95% CI 6.1–10.3) months. Survival rates at one and two years were 39.6% and 17.1%, respectively. Estimated incidence of local failure using competing risk analysis were 24% and 60% at 1 and 2 years, respectively. The overall response

**Funding:** The authors received no specific funding for this work.

**Competing interests:** The authors have declared that no competing interests exist.

rate was 74%, with an 18.5 percent complete response rate. In multivariate analysis, tumor size, overall response, and radiation dose were all significant prognostic variables for OS. Hepatic unfavorable events of grade 3 and 4 were for 14.1% of the total. There was no occurrences of grade 3–4 gastrointestinal toxicity, either acute or late. Additionally, there were no treatment-related mortality.

## Conclusions

Advanced RT is regarded as a safe and effective therapeutic option for HCC with PVTT. Overall survival was clearly related to tumor size, radiation dose, and tumor/PVTT response. Individuals with BED 56 $Gy_{10}$ had significantly better overall survival than patients with BED 56 $Gy_{10}$. A prospective randomized trial is required to validate these outcomes in order to corroborate these findings.

## Introduction

Hepatocellular carcinoma is the sixth most common cancer in the world (HCC). It is also the fourth leading cause of cancer death [1]. In patients with advanced-stage HCC, portal vein thrombosis (PVTT) is a poor prognostic factor. PVTT disrupts the normal liver's blood supply, resulting in a variety of significant consequences such as portal venous hypertension, hematemesis, and ascites. Furthermore, PVTT may have a role in intrahepatic and distant metastases [2]. When the patients did not get any treatment, their median survival was only 2.4 months [3].

The multi-tyrosine kinase inhibitors (TKI) are the standard treatment for locally advanced or metastatic HCC based on the prospective randomized phase III trials. However, the median survival benefit was only 2.5 months, as compared with those who received a placebo [4,5]. In addition, the tumor response rate was only 2–3%, and the median survival was only 8.1 and 5.6 months in the subgroup of HCC patients with vascular invasion in the SHARP and Pacific study, respectively. Moreover, the median time to progression was only 2.8 months in a study conducted in the Asia-Pacific region [4,5].

Historically, the efficacy of radiotherapy in liver cancer has been restricted by low liver tolerance to whole-liver irradiation. Radiation-induced liver disease (RILD), which can lead to liver failure and death, is the most serious issue and dose-limiting toxicity of liver radiation [6]. As a result, RT had a small role in the treatment of liver cancers. Recent developments in radiation planning and delivery technology, on the other hand, have enabled the safe delivery of higher and more effective doses. RT has been shown to be effective in the treatment of HCC with vascular invasion [7–18]. However, research on sophisticated RT techniques is currently limited.

The purpose of this study was to examine the clinical results and toxicity of advanced RT method in patients with HCC who had PVTT.

## Materials and methods

### Ethical consideration

The Chulalongkorn University Faculty of Medicine's Research Ethics Committee approved the retrospective analysis of all cases included in this study. No written consent was obtained in accordance with local ethical standards. As a result, the ethics committee waived the informed consent requirement. Prior to data collection, all data were anonymized.

## Patients

From July 2007 to August 2019, 160 patients with HCC and PVTT who had radiotherapy at King Chulalongkorn Memorial Hospital (KCMH) in Bangkok, Thailand, were included in the study. HCC was diagnosed based on one or more of the following criteria: 1) pathologically confirmed; 2) at least one solid liver lesion or vascular tumor thrombosis > 1 cm on multiphasic computed tomography (CT) or magnetic resonance imaging (MRI) in the presence of cirrhosis or chronic hepatitis B or C without cirrhosis. PVTT was identified by the existence of a low attenuation intraluminal filling defect during the portal phase and a filling defect enhancement during the arterial phase. This study excluded patients with extrahepatic metastases (M1). The Eastern Cooperative Oncology Group (ECOG-PS) criteria were used to assess performance status.

## Treatment consideration

The treatment choice in HCC patients with PVTT of the surgeon, hepatologist, interventionist, and radiation oncologist was made using multi-disciplinary approaches as the institute's standard of care. All treatments have been clinically verified and supported by evidence. The physician has designed the sequence of therapies to be the best appropriate for each patient's condition.

## Radiotherapy

Patients were positioned supine with both arms raised above the head for multiphasic CT simulation. The exhale breath-hold CT was used as the baseline for RT planning. If breath-hold scanning was not possible, free-breathing or average phase CT (from 4D CT) could be used. The information was entered into the Eclipse planning system (Varian, Palo Alto, CA).

The primary tumor, PVTT, normal liver, kidneys, small bowel, stomach, and spinal cord were contoured. The gross tumor volume (GTV) was conventionally defined as the total volume of PVTT and parenchymal HCC. However, in patients with large tumors and severe liver cirrhosis or numerous intrahepatic metastasis, only the PVTT was delineated as the GTV. There was no expansion from GTV to the clinical target volume (CTV) for most cases. A planning target volume (PTV) margin around CTV depended on the motion management that used the patients' motion and reproducibility. PTV margins of 0.5–1.0 cm were used for most patients with exhale-breath hold. If free breathing was used, the cranial-caudal margins were 1.5 cm, and the radial margins were 1.0 cm.

Radiation techniques included 3D-conformal radiotherapy (3D-CRT), intensity-modulated radiotherapy (IMRT), volumetric-modulated radiotherapy (VMAT), and stereotactic body radiotherapy (SBRT) using a linear accelerator with a 6–15 MV output and a flattening filter-free beam. The radiation dose was 25–50 Gy in 4–5 fractions in the SBRT group. Patients in the 3D-CRT, IMRT, and VMAT groups received either a hypofractionation schedule of 30–55 Gy in 10–15 fractions or a conventional dose of 45–60 Gy in 25–30 fractions. The plan was optimized based on tumor size, location, and normal liver volume. The goal was to use the highest prescription dose to the target volume and the shortest treatment duration, while adhering to normal tissue constraints.

## Post-treatment evaluation

PVTT and primary tumor response were assessed 2 to 3 months after RT with a multiphasic CT scan or MRI. To evaluate therapeutic response, the modified Reviewed Response Evaluation Criteria in Solid Tumors (mRECIST) criteria were utilized. The disappearance of arterial

enhancement was classified as a complete response (CR), at least a 30% decrease in arterial enhancement as a partial response (PR), an increase of at least 20% in arterial enhancement as progressive disease (PD), and no change or slight change that did not fulfill the criteria for PR or PD as stable disease (SD). PVTT was categorized a non-measurable lesion. Complete resolution of arterial enhancement was characterized as a complete response (CR), arterial enhancement persistence as an incomplete response/stable disease, and unequivocal progression as progressive disease (PD). The outcome of a combined primary tumor assessment and PVTT response was defined as the overall response.

Patients were assessed weekly during the RT period, and subsequently at 1- and 4-months post-treatment for liver and gastrointestinal toxicity. The Common Toxicity Criteria for Adverse Events (CTCAE) version 5.0 was used for grading.

## Outcomes and statistical analyses

The primary outcome, overall survival (OS), was defined from the first day of the RT to the time of death. Secondary outcomes included the response rate of the main tumor and PVTT, local failure, acute toxicity, late adverse events of the liver and gastrointestinal tract, and variables related with OS.

The overall survival rate was estimated using the Kaplan-Meier method. The factors influencing the survival rate were determined using cox's regression analysis, in which significant variables ($p < 0.2$) in univariate analysis were included in a multivariate analysis.

Local failure was defined as a new nodular or irregular enhancing lesion on CT scan in the previously treated PTV region, as well as a nodular or irregular lesion on MRI with typical malignant tumor tissue signal characteristics.

Furthermore, the cumulative incidence of local recurrence was computed and compared across groups using the Gray's test, with death as a competing risk.

To enable for dose comparison, a biological equivalent dose (BED) was derived using the formula BED (Gy) = dose/fraction x fraction number (1 + fraction dose/ $\alpha/\beta$), with $\alpha/\beta$ of 10 Gy for tumor tissue.

The response rate and toxicity were summarized using descriptive statistics. The Pearson's chi-square test was used to compare categorical variables.

SPSS 22.0 and Stata 15.0 were used for all calculations. A P value of less than 0.05 was considered statistically significant.

## Results

### Baseline characteristics

Between July 2007 and August 2019, 160 HCC with PVTT patients received RT in our institution. Patient characteristics are shown in Table 1. The mean age of the patients was 60.8 (95% CI 58.2–62.0) years old. The large percentage of the patients (88.1%) were male. Most of the patients (76.9%) had an ECOG performance status of 0 or 1. Chronic hepatitis infection was found in 71.3% of the patients. One hundred and nine patients (69.4%) had Child-Pugh A at the time of radiation. The median tumor size was 7.7 (range 1–24.5) cm. Tumor thrombosis involved the main trunk or bilateral first branch in 85 patients (54.5%), the unilateral first branch in 51 patients (32.7%), and IVC in 20 patients (12.5%).

In 2015, our department installed sophisticated SBRT machines. From 2015 to 2019, 120 patients (75 percent) were treated. The median size of tumors and the proportion of patients with tumors larger than 10 cm were greater in 2015 and subsequent years. Since 2015, the IMRT/VMAT technology has been more frequently used. However, the tumor BED was not different in the two groups (S1 Table).

**Table 1. Patient and treatment characteristics.**

| Variables | Number of patients (%) |
|---|---|
| Age (n = 160) (mean ± SD) | 60.8 ± 12.1 |
| Sex | |
| Male | 19 (11.9%) |
| Female | 141 (88.1%) |
| ECOG-PS | |
| 0–1 | 123 (76.9%) |
| 2 | 37 (23.1%) |
| Child-Pugh score | |
| Median (range) | 6 (5–10) |
| 5–6 | 109 (69.4%) |
| 7–9 | 45 (28.7%) |
| 10–15 | 3 (1.9%) |
| Missing | 3 (1.9%) |
| Underlying liver disease | |
| HBV infection | 75 (46.9%) |
| HCV infection | 39 (24.4%) |
| Tumor size (cm) | |
| Median (range) | 7.7 (1.0–24.5) |
| <10 cm | 94 (59.1%) |
| ≥10cm | 65 (40.9%) |
| Missing | 1 (0.6%) |
| Presence of tumor thrombus | |
| IVC | 20 (12.5%) |
| Portal vein | 156 (97.5%) |
| Site of PVTT | |
| Main or bilateral portal vein | 85 (54.5%) |
| Unilateral portal vein | 51 (32.7%) |
| Others | 20 (12.8%) |
| Missing | 4 (2.5%) |
| BCLC stage | |
| C | 157 (98.1%) |
| D | 3 (1.9%) |
| TNM stage (AJCC 8th) | |
| T stage | |
| T1a | 1 (0.6%) |
| T1b | 0 (0.0%) |
| T2 | 12 (7.5%) |
| T3 | 6 (3.8%) |
| T4 | 141 (88.1%) |
| N stage | |
| N0 | 142 (88.8%) |
| N1 | 18 (11.3%) |
| M stage | |
| M0 | 160 (100%) |
| Treatment timing | |
| 2009–2014 | 40 (25%) |
| 2015–2019 | 120 (75%) |

(*Continued*)

**Table 1.** (Continued)

| Variables | Number of patients (%) |
|---|---|
| Radiation technique | |
| 3D-CRT | 40 (25.0%) |
| IMRT/VMAT | 100 (62.5%) |
| SBRT | 20 (12.5%) |
| Fractionation | |
| Conventional | 23 (14.4%) |
| Hypofractionation | 117 (73.1%) |
| SBRT | 20 (12.5%) |
| BED ($Gy_{10}$) | |
| Mean ± SD | 49.60 ± 18.63 |
| < 56 $Gy_{10}$ | 98 (61.2%) |
| ≥ 56 $Gy_{10}$ | 62 (38.8%) |
| GTV volume (cc) | |
| Median (range) | 309.0 (5.5–4774.0) |
| Previous treatment | |
| Surgery | 15 (9.4%) |
| TACE | 95 (59.4%) |
| Embolization | 4 (2.5%) |
| Chemotherapy | 5 (3.1%) |
| Sorafenib | 7 (4.4%) |
| MVA | 3 (1.9%) |
| RFA | 8 (5.0%) |
| Y-90 | 3 (1.9%) |
| Post RT treatment | |
| Surgery | 0 (0%) |
| TACE | 43 (26.9%) |
| Embolization | 2 (1.3%) |
| Chemotherapy | 5 (3.1%) |
| Sorafenib | 12 (7.5%) |
| MVA | 2 (1.3%) |
| RFA | 7 (4.4%) |
| Y-90 | 3 (1.9%) |
| Others | 5 (3.1%) |

Radiation techniques were mostly IMRT/VMAT (62.5%). The hypofractionation scheme was adopted in 73.1%. And SBRT was used in 12.5% of patients. The median GTV volume was 309.0 (range 5.5–4774.0) cc. The mean biological equivalent dose (BED) as α/β ratio of 10 was 49.6 (Range 14.4–100) $Gy_{10}$. The biological effective equivalent of 2-Gy fractions, as α/β ratio of 10 (EQD2), were also calculated, and the mean EQD2 was 41.3 (Range 12–83.3) $Gy_{10}$. The mean of mean liver dose in conventional, hypofractionated, and SBRT groups were 21 ± 6.6, 15 ± 4.8, and 12 ± 3.7 Gy, respectively.

One hundred nine patients (68.1%) had received at least one treatment prior to radiation. Ninety-five patients (59.4%) had previously treated with TACE, seven patients (4.4%) received sorafenib, and Y-90 in 3 patients (1.9%). In terms of post RT treatment, TACE was undergone in 43 patients (26.9%), sorafenib in 12 patients (7.5%), and Y-90 in 3 patients (1.9%).

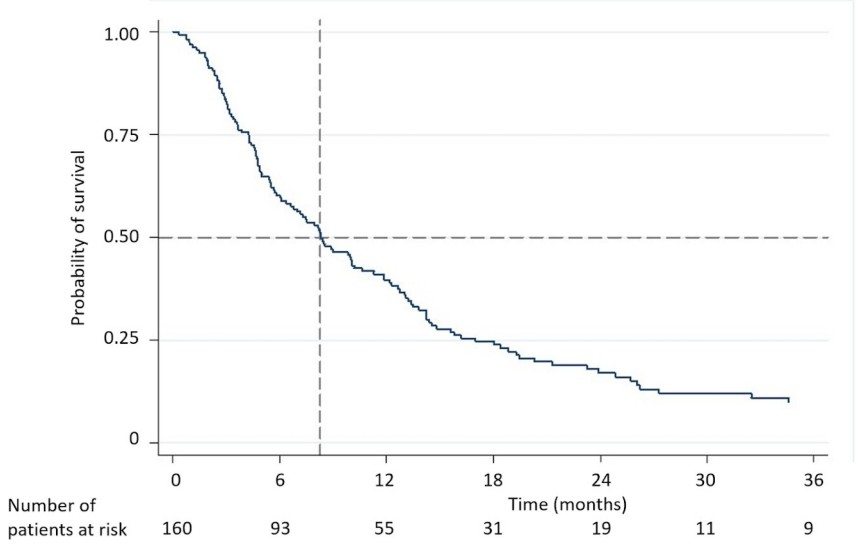

**Fig 1. Kaplan-Meier analysis of overall survival.**

## Overall survival and factors related to OS

The median follow-up time was 8.2 (95% CI 9.8–14.6) months. The overall survival rates at 1, 2, and 3 years were 39.6%, 17.1%, and 9.9%, with the median survival of 8.3 (95% CI 6.1–10.3) months (Fig 1).

In the univariate analysis, age $\geq$ 60 years, ECOG PS of 0 or 1, pretreatment Child-Pugh A, small tumor size (< 10 cm), higher radiation dose (BED $\geq$ 56 $Gy_{10}$), and RT responder (CR, PR) were favorable prognostic factors of survival. Multivariate analysis demonstrated that tumor size, radiation dose, and overall response were all significant independent predictors of OS (Table 2).

Patients with tumor size < 10 cm had higher overall survival than $\geq$ 10 cm The median survival in was 12.7 months in tumor size < 10 cm as compared to 4.8 months in tumor size $\geq$ 10 cm. Survival rates at one and two years were 52.0 percent and 25.8 percent, respectively, in the tumor size < 10 cm group, and 20.2 percent and 0% in the tumor size $\geq$ 10 cm group (p < 0.001) (Fig 2).

Patients who received the higher radiation dose (BED $\geq$ 56 $Gy_{10}$) had a considerably better overall survival rate than those who received the lower radiation dose (BED < 56 $Gy_{10}$). The

**Table 2. Univariate and multivariate analyses of potential prognostic factors for overall survival.**

| Variables | Univariate analysis | | | Multivariate analysis | | |
|---|---|---|---|---|---|---|
| | HR | 95% CI | P-value | HR | 95% CI | P-value |
| **Age (< 60 yr vs $\geq$ 60 yr)** | 1.35 | 0.96–1.91 | 0.09 | 1.03 | 0.66–1.60 | 0.90 |
| **ECOG-PS (2 vs 0–1)** | 2.08 | 1.39–3.09 | < 0.001 | 1.41 | 0.79–2.52 | 0.24 |
| **Child-Pugh score (B or C vs A)** | 1.85 | 1.28–2.67 | 0.001 | 1.07 | 0.64–1.79 | 0.79 |
| **Hepatitis B/C infection (yes vs no)** | 1.02 | 0.69–1.49 | 0.94 | | | |
| **Tumor size ($\geq$ 10 cm vs < 10 cm)** | 2.62 | 1.82–3.77 | < 0.001 | 2.00 | 1.27–3.14 | 0.003 |
| **PVTT location (main/bilateral first branch vs others)** | 1.22 | 0.86–1.73 | 0.26 | | | |
| **BED10 (< 56 Gy vs $\geq$ 56 Gy)** | 2.64 | 1.81–3.83 | < 0.001 | 1.83 | 1.15–2.90 | 0.01 |
| **Overall response (SD/PD vs CR/PR)** | 2.04 | 1.31–3.19 | 0.002 | 2.00 | 1.26–3.17 | 0.003 |
| **RT technique (non-SBRT vs SBRT)** | 1.18 | 0.71–1.96 | 0.53 | | | |

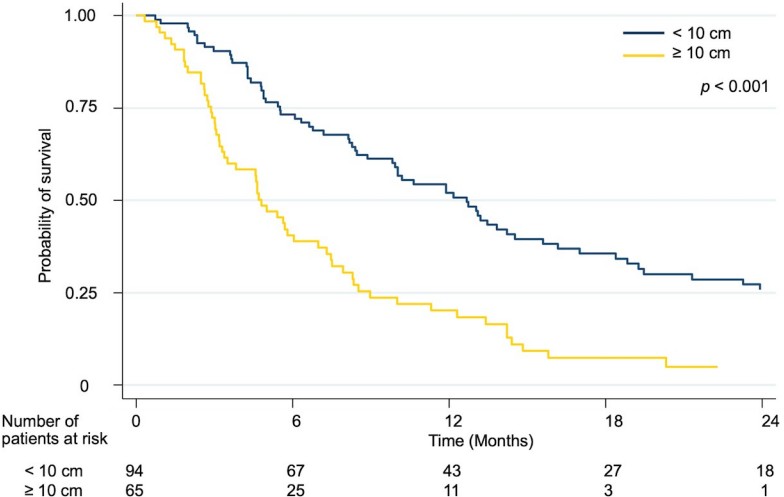

**Fig 2. Kaplan-Meier curves for overall survival rate according to the tumor size.**

median survival, 1-year and 2-year OS rate were 14.4 (95% CI 10.5–18.3) months, 66.7% and 28.1% in BED $\geq$ 56 Gy$_{10}$ group and 5.0 (95% CI 3.9–6.1) months, 22.4% and 10.3% in BED < 56 Gy$_{10}$ group (p < 0.001) (Fig 3).

The median survival, one-year, and two-year OS rates were 13.1 (95% CI 11.5–14.7) months, 56.4%, and 26.5%, respectively, in responders, and 6.8 (95% CI 3.8–9.7) months, 25.8%, and 7.4%, respectively, in non-responders. Overall survival was significantly greater for responders than for non-responders (p = 0.002) (Fig 4).

## Competing risk analysis of local failure and associated factors

Estimated incidence of local failure using competing risk analysis were 24% and 60% at 1 and 2 years, respectively. (Fig 5) PVTT location (Main, bilateral, or first branch) and overall response (SD/PD) were unfavorable predictors for local recurrence in univariate analysis. Multivariate analysis revealed that the sole factor influencing local relapse is overall response.

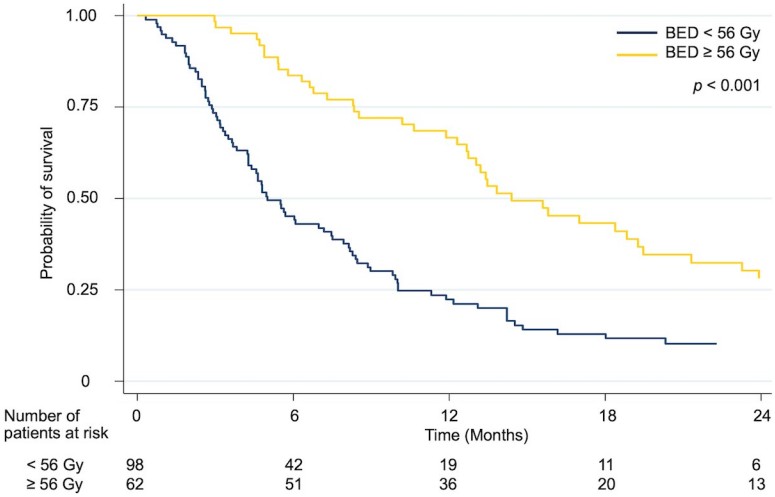

**Fig 3. Kaplan-Meier curves for overall survival rate according to the radiation dose.**

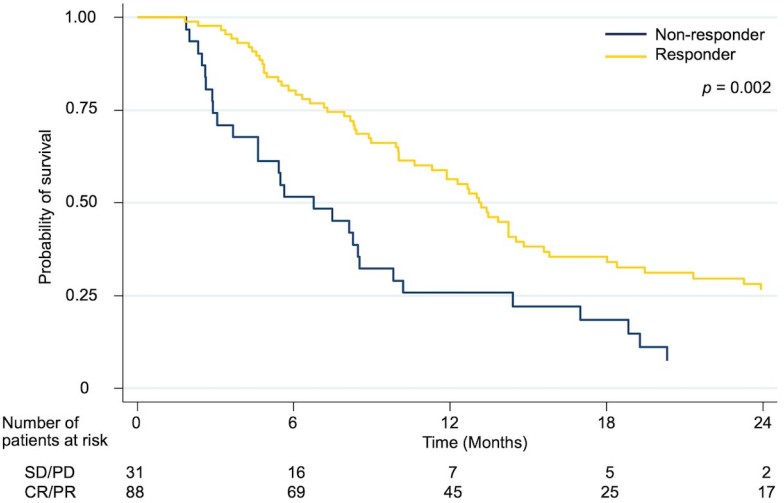

**Fig 4. Kaplan-Meier curves for overall survival rate according to the response after radiation.**

(Table 3) The estimated one-year and two-year cumulative local recurrence rates were 87 and 99 percent, respectively. In comparison, the responder group had a cumulative 1 and 2 year local relapse rate of 22% and 43%, respectively (p<0.001) (Fig 6).

## Response rate

The response rates were summarized in Table 4. The tumor response, PVTT, and overall response could be assessed in 81, 117, and 119 patients, respectively. Tumor responses were: CR in 16 (19.8%) patients, PR in 44 (54.3%) patients, SD in 12 (14.8%) patients and PD in 9 (11.1%) patients. The response rate (CR + PR) of the primary tumor was 74.1%. PVTT responses were: CR in 26 (22.2%) patients, incomplete response or SD in 72 (61.5%) patients

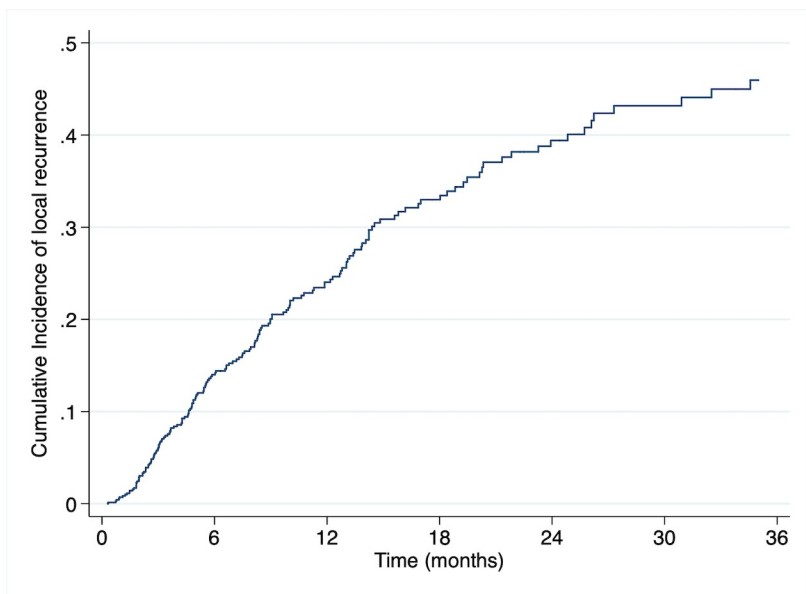

**Fig 5. Estimated cumulative incidence curves illustrating local recurrence by competing risk analysis (any cause of death as a competing risk event.**

**Table 3. Univariate and multivariate competing risk analysis of factors associated with local recurrence.**

| Variables | Univariate analysis | | | Multivariate analysis | | |
|---|---|---|---|---|---|---|
| | SHR | 95% CI | P-value | SHR | 95% CI | P-value |
| Tumor size ($\geq$ 10 cm vs < 10 cm) | 0.89 | 0.47–1.67 | 0.71 | | | |
| PVTT location (main/bilateral first branch vs others) | 2.76 | 0.67–11.24 | 0.16 | 2.99 | 0.70–12.80 | 0.14 |
| BED10 (< 56 Gy vs $\geq$ 56 Gy) | 1.00 | 0.56–1.79 | 0.99 | | | |
| Overall response (SD/PD vs CR/PR) | 6.86 | 3.49–13.47 | < 0.001 | 7.51 | 3.69–15.26 | < 0.001 |
| RT technique (non-SBRT vs SBRT) | 1.01 | 0.37–2.79 | 0.98 | | | |

and PD in 19 (16.2%) patients. Overall responses were: CR in 22 (18.5%) patients, PR in 66 (55.5%) patients, SD in 10 (8.4%) patients and PD in 21 (17.6%) patients. The overall response rate (CR + PR) was 74.0%. The overall responder was significantly higher in the higher dose group (BED $\geq$ 56 $Gy_{10}$) than in the lower dose group (82.8% vs 65.6%, respectively; p = 0.04) (Table 5).

## Toxicity

The acute liver toxicities were shown in Table 6. No treatment-related grade 5 acute toxicity was found within 4 months after RT. Grade 3/4 hepatic toxicities were found 14.1%. Univariate analysis for predictive factors related to grade 3/4 adverse events has been evaluated. There was no correlation between predictive factors. (S2 Table) The majority of gastrointestinal side effects occurred during radiation treatment. Vomiting of grade 1 and 2 was reported in 2.5 percent and 1.3 percent of the participants, respectively. There was no evidence of late gastrointestinal morbidity, such as perforation, stenosis, or ulceration.

## Discussion

Due to concerns about RILD, RT has historically had a minor role in the management of HCC. With the advancement of RT technology, it has been increasingly used in HCC patients

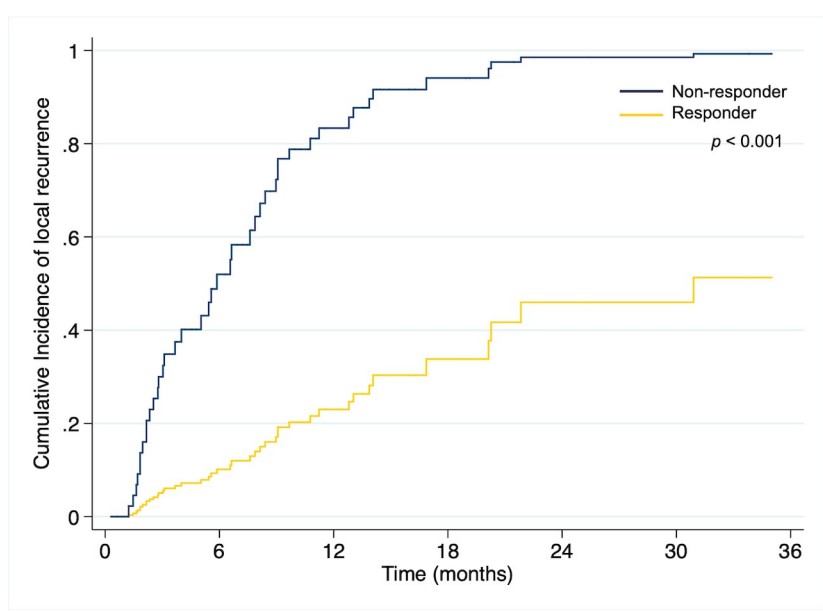

**Fig 6. Competing risk analysis of local failure depending on overall response.**

**Table 4. Tumor, PVTT and overall response after the completion of the RT.**

| Response | Tumor Response (n = 81) | PVTT Response (n = 117) | Overall Response (n = 119) |
|---|---|---|---|
| CR | 16 (19.8%) | 26 (22.2%) | 22 (18.5%) |
| PR | 44 (54.3%) | 72 (61.5%) | 66 (55.5%) |
| SD | 12 (14.8%) | | 10 (8.4%) |
| PD | 9 (11.1%) | 19 (16.2%) | 21 (17.6%) |
| Responder (CR + PR) | 60 (74.1%) | 98 (83.8%) | 88 (74.0%) |
| Non-responder (SD + PD) | 21 (25.9%) | 19 (16.2%) | 31 (26.0%) |

in recent years. Several modest series demonstrated that RT resulted in a favorable response and promising treatment outcomes for patients who were considered unsuitable for other modalities. A randomized phase II study compared TACE with RT to sorafenib in HCC with PVTT patients showed a positive effect of TACE-RT. The TACE-RT group showed a significantly higher 12-week progression-free survival (86.7% vs 34.3%; P < 0.001), significantly longer overall survival (MS 55.0 vs 43.0 weeks; P = 0.04) and significantly higher response rate (33.3% vs 2.2%; P < 0.001) [19]. Moreover, two retrospective studies also showed that overall survival was significantly higher in the RT group when compared to the sorafenib group in a propensity score-matched cohort [20,21]. These findings indicate that local RT should be considered a treatment option for HCC with PVTT patients.

In previous studies, the median OS in patients with HCC with PVTT treated with RT was 5.3–17.0 months. In our series, the overall survival rate was 8.3 months, which was not different from other reports (Table 7) [7–13,16,18]. Among the studies, the population was heterogeneous. Almost all patients in this study had major portal vein and IVC thrombosis, which were considered the worse prognosis group.

In multivariate analysis, tumor size, overall response to RT and radiation dose were significant independent prognostic factors for overall survival. Responders lived longer with a median overall survival of 13.1 months compared to non-responders with a median overall survival of 6.8 months. Several studies suggested that overall survival was statistically higher for those who responded to radiation. The median OS in responders and non-responder was 9.7–22 months and 3.8–7.2 months, respectively [7,9,10,13,15,16].

Our study found a 74.0% objective response that appears to be higher than the previous studies. The differences between the objective response rates reported in this study and rates reported previously [7–13,16,18] might be due to the differences in terms of response criteria between studies. The mRECIST was used to evaluate response in this study because several clinical investigations have shown it predicts survival in patients receiving loco-regional therapies better than conventional methods (RECIST) [22]. Moreover, the recent trials, which evaluated the efficacy of treatments in unresectable hepatocellular carcinoma, were based on these criteria to assess the responses. [4,5,23,24] Several studies suggested that higher radiation doses could improve tumor response and overall survival (Table 8). Kim et al. [13] found that PVTT objective response rates were 20% and 54.6% in patients treated with BED of $< 58$ $Gy_{10}$ and $\geq 58$ $Gy_{10}$. The responder had a median survival of 10.7 months, while non-responders

**Table 5. Relationship between RT dose and overall response rate.**

| Overall Response (%) | RT dose (BED) | | P-value |
|---|---|---|---|
| | $< 56$ $Gy_{10}$ | $\geq 56$ $Gy_{10}$ | |
| Responder (CR + PR) | 40 (65.6%) | 48 (82.8%) | 0.038 |
| Non-responder (SD + PD) | 21 (34.4%) | 10 (17.2%) | |

**Table 6. Acute hepatic toxicities after RT.**

| CTCAE 5.0 | Grade 0 | Grade 1 | Grade 2 | Grade 3 | Grade 4 | Grade 5 |
|---|---|---|---|---|---|---|
| AST (n = 155) | 74 (47.7%) | 55 (35.5%) | 9 (5.8%) | 13 (8.4%) | 4 (2.6%) | - |
| ALT (n = 154) | 79 (51.3%) | 54 (35.1%) | 8 (5.2%) | 10 (6.5%) | 3 (1.9%) | - |
| ALP (n = 154) | 115 (74.7%) | 34 (22.1%) | 5 (3.2%) | 0 (0%) | 0 (0%) | - |
| TB (n = 156) | 57 (36.5%) | 32 (20.5%) | 45 (28.8%) | 16 (10.3%) | 6 (3.8%) | - |
| Alb (n = 143) | 9 (6.3%) | 68 (47.6%) | 52 (36.4%) | 14 (9.8%) | 0 (0%) | 0 (0%) |
| INR (n = 141) | 60 (42.6%) | 61 (43.3%) | 18 (12.8%) | 2 (1.4%) | 0 (0%) | - |

had a median survival of 5.3 months (p = 0.05). Toya et al. [8] also found a correlation between BED and not only PVTT response but also OS. The response rate and 1-year survival were higher in patients who received BED $\geq$ 58 $Gy_{10}$ rather than < 58 $Gy_{10}$ (80% vs 21.7% and 59.3% vs 29.2%, respectively). Kim et al. [14] found that there was no objective response of PVTT in patients receiving a BED dose < 64 $Gy_{10}$, but there was a 50% response of PVTT in patients receiving a BED dose $\geq$ 64 $Gy_{10}$ (p < 0.007). In HCC with PVTT patients, the responder lived longer than non-responders (MS 20.1 vs 7.2 months, p < 0.007). These results are similar to our study, in which a BED dose $\geq$ 56$Gy_{10}$ had a response rate of 82.8%, while a BED < 56 $Gy_{10}$ had a response rate of 65.6% (p = 0.04). Additionally, there was a median survival of 14.4 months in patients receiving a BED dose $\geq$ 56 $Gy_{10}$ while patients who were receiving a BED < 56 $Gy_{10}$ had a median survival of 5.0 months (p < 0.001).

Several studies suggested that higher radiation doses could improve tumor response and overall survival (Table 8). Kim et al. [13] found that PVTT objective response rates were 20% and 54.6% in patients treated with BED of < 58 $Gy_{10}$ and $\geq$ 58 $Gy_{10}$. The responder had a median survival of 10.7 months, while non-responders had a median survival of 5.3 months (p = 0.05). Toya et al. [8] also found a correlation between BED and not only PVTT response but also OS. The response rate and 1-year survival were higher in patients who received BED $\geq$ 58 $Gy_{10}$ rather than < 58 $Gy_{10}$ (80% vs 21.7% and 59.3% vs 29.2%, respectively). Kim et al. [14] found that there was no objective response of PVTT in patients receiving a BED dose < 64 $Gy_{10}$, but there was a 50% response of PVTT in patients receiving a BED dose $\geq$ 64 $Gy_{10}$ (p < 0.007). In HCC with PVTT patients, the responder lived longer than non-responders (MS 20.1 vs 7.2 months, p < 0.007). These results are similar to our study, in which a BED

**Table 7. The median survival and objective response rate of prospective and retrospective studies applying radiotherapy in HCC with PVTT.**

| Studies | Year | Design | Treatment | No. | OR | Median survival (months) | | | |
|---|---|---|---|---|---|---|---|---|---|
| | | | | | | All | Responder | Non-responder | P value |
| Tazawa et al. [10] | 2001 | Retrospective | TACE + RT | 24 | 50% | 12.5 (A) 2.7 (B) | 9.7 | 3.8 | < 0.001 |
| Ishikura et al [11] | 2002 | Prospective | TACE + RT | 20 | 50% | 5.3 | NA | NA | NA |
| Yamada et al. [12] | 2003 | Prospective | TACE + RT | 19 | 57.9% | 7 | 15.4 | 4.6 | 0.16 |
| Kim et al. [13] | 2005 | Retrospective | RT | 59 | 45.8% | 10.7 | 10.7 | 5.3 | 0.05 |
| Nakazawa et al. [15] | 2007 | Retrospective | RT | 32 | 48% | 5.7 | 13.8 | 7.0 | 0.01 |
| Toya et al. [8] | 2007 | Retrospective | RT | 38 | 44.7% | 9.6 | NA | NA | NA |
| Zeng et al. [7] | 2008 | Retrospective | RT | 136 | 57.6% | 9.7 | 19.5(CR) 10.2(PR) | 7.2 (SD) 3.5 (PD) | < 0.001 |
| Yu et al. [9] | 2011 | Retrospective | RT ± TACE | 281 | 53.8% | 11.6 | 22.0 | 5.0 | < 0.001 |
| Yoon et al. [16] | 2012 | Retrospective | TACE + RT | 412 | 39.6% | 10.6 | 19.4 | 7.0 | < 0.001 |
| Kim et al. [18] | 2014 | Retrospective | TACE + RT | 59 | 51.0% | 17 | NA | NA | NA |
| This study | 2021 | Retrospective | RT | 160 | 74.0% | 8.3 | 13.1 | 6.8 | 0.002 |

**Table 8. The correlation between radiation dose, response rate and overall survival.**

| Studies | N | Relationship | Outcomes | P-value |
|---|---|---|---|---|
| **Kim et al. [13]** | 59 | BED and PVTT response | BED $< 58Gy_{10}$ Responder 20%<br>BED $\geq 58Gy_{10}$ Responder 54.6% | 0.034 |
| | | PVTT response and OS | Responder MS 10.7 mo<br>Non-responder MS 5.3 mo | 0.050 |
| **Toya et al. [8]** | 38 | BED and PVTT response | BED $< 58Gy_{10}$ Responder 21.7%<br>BED $\geq 58Gy_{10}$ Responder 80% | 0.0007 |
| | | BED and OS | BED $< 58Gy_{10}$ 1-y OS 29.2%<br>BED $\geq 58Gy_{10}$ 1-y OS 59.3% | 0.0421 |
| **Kim et al. [14]** | 70 (PVTT 41) | BED and PVTT response | BED $< 64Gy_{10}$ Responder 0%<br>BED $\geq 64Gy_{10}$ Responder 50% | $< 0.007$ |
| | | PVTT response and OS | Responder MS 20.1 mo<br>Non-responder MS 7.2 mo | 0.007 |
| **This study** | 160 | BED and overall response | BED $< 56Gy_{10}$ Responder 65.6%<br>BED $\geq 56Gy_{10}$ Responder 82.8% | 0.038 |
| | | BED and OS | BED $< 56Gy_{10}$ MS 5 months<br>BED $\geq 56Gy_{10}$ MS 14.4 months | $< 0.001$ |

dose $\geq 56Gy_{10}$ had a response rate of 82.8%, while a BED $< 56$ $Gy_{10}$ had a response rate of 65.6% (p = 0.04). Additionally, there was a median survival of 14.4 months in patients receiving a BED dose $\geq 56$ $Gy_{10}$ while patients who were receiving a BED $< 56$ $Gy_{10}$ had a median survival of 5.0 months (p $< 0.001$).

There were various limitations to assess local failure in hepatocellular cancer with PVTT. The patient may have undergone a combination of treatments before or after radiation. Furthermore, at PVTT, the criteria for identifying local relapse are not explicitly defined. This is a shortcoming of our research as well.

SBRT is the most effective technique. Concentrating on the SBRT subgroup within our cohort, our research included the treatment of twenty patients with SBRT. The median tumor size in this cohort was 3.9 cm (range 1.7–18.4) The mean BED dose in this group was 75.9 Gy. (S3 Table) We observed no difference in overall survival between the SBRT and non-SBRT groups. (S4 Table and S1 Fig) It is likely that the sample size and power of this group are insufficient to demonstrate a significant difference.

SBRT is not extensively used in HCC with PVTT. These retrospective investigations revealed an overall response rate of 54.1–87 percent, resulting in overall survival ranged between 10 and 20.8 months [25–29]. Our study exhibited a similar outcome of 80% overall response rate and 11.9 months median overall survival (S5 Table).

Grade 3/4 hepatic toxicities were observed in 14.1% of patients. Likewise, the toxicities were acceptable and no fatal complication. These hepatic adverse effects were usually self-limited with supportive care. This study found no correlation between factors in grade 3/4 hepatic toxicities. It is possibly from the limited number of events. Furthermore, the RT prescription doses were restricted by hard constraints of mean liver dose.

The study has several limitations. First, due to the retrospective nature of our study, there was a risk of selection bias. Second, there were variations of radiation treatments and baseline patient characteristics. Third, the response of tumor and PVTT after radiation was difficult to evaluate and post-RT treatment could contribute to the survival effects in this study.

## Conclusions

RT is now considered a feasible, safe and effective treatment option for HCC with PVTT. The tumor size, radiation dose, and response of the primary tumor and PVTT were clearly related

to overall survival. Overall survival was significantly higher in patients with BED $\geq 56$ $Gy_{10}$ compared to patients with BED $< 56$ $Gy_{10}$), but a prospective randomized study is needed to confirm these results

## Supporting information

**S1 Fig. Kaplan-Meier curves for overall survival rate according to the radiation technique.** (TIF)

**S1 Table. Comparison of patient and treatment characteristics between 2007 to 2014 and 2015 to 2019.** (DOCX)

**S2 Table. Univariate analysis of potential predictive factors for any grade 3/4 liver toxicities.** (DOCX)

**S3 Table. Comparison of patient and treatment characteristics between SBRT and Non-SBRT groups.** (DOCX)

**S4 Table. Survival data comparison between SBRT and Non-SBRT groups.** (DOCX)

**S5 Table. The median survival time and objective response rate in studies employing SBRT in patients with HCC with PVTT.** (DOCX)

## Acknowledgments

First, we thank Dr. Wirote Lausoontornsiri for rechecking. We thank Miss Buntipa Netsawang for information and partly statistical assistance. Moreover, we thank the medical physicist team at King Chulalongkorn Memorial Hospital for technical and treatment planning facilitation.

## Author Contributions

**Conceptualization:** Chonlakiet Khorprasert, Petch Alisanant, Napapat Amornwichet.

**Data curation:** Kanokphorn Thonglert, Napapat Amornwichet.

**Formal analysis:** Chonlakiet Khorprasert, Kanokphorn Thonglert.

**Investigation:** Chonlakiet Khorprasert, Kanokphorn Thonglert, Napapat Amornwichet.

**Methodology:** Chonlakiet Khorprasert, Napapat Amornwichet.

**Project administration:** Napapat Amornwichet.

**Resources:** Petch Alisanant.

**Supervision:** Chonlakiet Khorprasert, Napapat Amornwichet.

**Validation:** Petch Alisanant, Napapat Amornwichet.

**Visualization:** Petch Alisanant.

**Writing – original draft:** Chonlakiet Khorprasert, Kanokphorn Thonglert, Napapat Amornwichet.

**Writing – review & editing:** Chonlakiet Khorprasert, Kanokphorn Thonglert, Petch Alisanant, Napapat Amornwichet.

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
