## [Decision Letter · Decision Letter 0]

18 May 2021

PONE-D-21-07678

Advanced radiotherapy technique in hepatocellular carcinoma with portal vein thrombosis: Feasibility and clinical outcomes

PLOS ONE

Dear Dr. Amornwichet,

Thank you for submitting your manuscript to PLOS ONE. After careful consideration, we feel that it has merit but does not fully meet PLOS ONE’s publication criteria as it currently stands. Therefore, we invite you to submit a revised version of the manuscript that addresses the points raised during the review process.

We look forward to receiving your revised manuscript.

Kind regards,

Panagiotis Balermpas

Academic Editor

PLOS ONE

Additional Editor Comments:

Dear authors,

your manuscript is very interesting,

however you will have to address the issues raised by the reviewers, especially the limitations and the discussion of the novelty/importance of your work, before the manuscript is suitable for publication.

Journal Requirements:

2. In your ethics statement in the manuscript and in the online submission form, please ensure that you have discussed whether all data/samples were fully anonymized before you accessed them and/or whether the IRB or ethics committee waived the requirement for informed consent. If patients provided informed written consent to have data/samples from their medical records used in research, please include this information.

4. Please include a copy of Tables 5-7 which you refer to in your text on pages 12-14.

Reviewers' comments:

Reviewer's Responses to Questions

**Comments to the Author**

1. Is the manuscript technically sound, and do the data support the conclusions?

Reviewer #1: Yes

Reviewer #2: Partly

2. Has the statistical analysis been performed appropriately and rigorously? 

Reviewer #1: Yes

Reviewer #2: No

3. Have the authors made all data underlying the findings in their manuscript fully available?

Reviewer #1: No

Reviewer #2: Yes

4. Is the manuscript presented in an intelligible fashion and written in standard English?

Reviewer #1: No

Reviewer #2: Yes

5. Review Comments to the Author

Reviewer #1: This is a well written and large retrospective overview of HCC/PVTT patients treated with radiotherapy. However, it does have some grammatical failures that should be corrected. Radiotherapy plays an increasingly important role in these patients and is included in the international guidelines as treatment option for Child Pugh A and B patients. SBRT appears to be even better than conventionally fractionated radiotherapy.

I have some small remarks:

Results:

Patients who were treated between 2007 and 2019 were included. Radiotherapy has improved a lot in this period. Did you see any differences in local control, radiotherapy type and OS between patients treated earlier vs. later?

Please correct EQD210 to EQD210

Please add the p value (site 10) to median survival comparing tumor size, also add the p value to survival comparing BED (site 11).

Did you also know the ECOG-PS of these patients?

Discussion:

The first paragraph is a copy of your introduction. Please remove in one of the 2 sections, or adapt.

Tables 5, 6 and 7 are not uploaded. Please add.

A part of your patients were treated with SBRT. Did you see a difference in LC and OS in this subgroup? Were these lesions smaller? Please add some more information on these SBRT patients in your results section.

Please compare your data to the newly published SBRT data in the discussion section.

Reviewer #2: The manuscript analyses a large HCC cohort with portal vein tumor thrombus retrospectively. As data on this HCC subpopulation are relatively rare, the database is valuable, however the manuscript and data analysis have major issues which have to be fixed before publication.

Major remarks:

- radiation techniques and different dose-fractionation methods were completely mixed up and this can not be completely compensated by just calculating BED. Please add subgroup analyses of normofractionated and hypo fractionated (SBRT) treatments

- what is meant by "advanced" RT technique? Techniques vary extremely from 3DCRT, IMRT, SBRT. Probably a subgroup analysis would bring more light to the data (eg group 1, SBRT; group 2, IMRT/VMAT/3DCRT with BED> a threshold and under this threshold

- statistical analysis: which software has been used? This info is missing from M&M

- please provide local control and competing-risk adjusted local. recurrence rates.

- toxicity part is very short. Please add non-hepatic toxicity (gastrointestinal, eg stomach, duodenum) which usually can happen by liver SBRT. How about central hepatobiliary toxicity, cholestasis, etc.

- table 1: some numbers do not seem to be correct/ do not add on to 100%.

there were 160 Patients, but some data are not provided for all of them eg CP-score only for 157, tumor size only for 159...).

For TNM, even 165 M0 stages are reported (but only 160 patients).

Please check, and provide information on missing data

- figure 2. probably the caption is mixed up. BED <10 cm?????

- figure 3. please add to the Kaplan Meier analysis a competing-risk adjusted local recurrence rate. probably we have. a survival time bias here with a lot of patients dying before local relapse.

Minor remarks:

- abstract: please explain "10"m do you mean alpa-beta ratio?

- do not begin a sentence with "and"

- it is not usual to name the treating center in the abstract, please remove

- follow-up instead of follows-up

-what is meant bei EQD210? probably EQD2- for alpha-beta 10?

6. PLOS authors have the option to publish the peer review history of their article (what does this mean?). If published, this will include your full peer review and any attached files.

Reviewer #1: No

Reviewer #2: No

---

## [Author Response · Author response to Decision Letter 0]

5 Jul 2021

Dear Editor,

 Thank you for inviting us to submit a revised draft of our manuscript entitled, "Advanced radiotherapy technique in Hepatocellular carcinoma with Portal Vein Thrombosis: Feasibility and clinical outcomes" to PLOS ONE. We also appreciate the time and effort you and each of the reviewers have dedicated to providing insightful feedback on ways to strengthen our paper. Thus, it is with great pleasure that we resubmit our article for further consideration. We have incorporated changes that reflect the detailed suggestions you have graciously provided. We also hope that our edits and the responses we provide below satisfactorily address all the issues and concerns you and the reviewers have noted.

To facilitate your review of our revisions, the following is a point-by-point response to the questions and comments delivered in your letter. All modifications in the manuscript have been highlighted in yellow.

EDITOR SUGGESTIONS:

• Thank you for your careful consideration. We have been thoroughly checked and corrected in accordance with PLOS ONE's style guidelines. 

2. In your ethics statement in the manuscript and in the online submission form, please ensure that you have discussed whether all data/samples were fully anonymized before you accessed them and/or whether the IRB or ethics committee waived the requirement for informed consent. If patients provided informed written consent to have data/samples from their medical records used in research, please include this information.

• As a response of your recommendation, we acknowledged your concern and addressed the ethical issue in the materials and methods section.

• As a response of your recommendation, we acknowledged your concern and addressed the ethical issue in the materials and methods section.

4. Please include a copy of Tables 5-7 which you refer to in your text on pages 12-14.

• We regret for the errors. They have been corrected.

• We adjusted the caption of the supplement material based on your recommendation.

REVIEWER 1 COMMENTS:

1. This is a well written and large retrospective overview of HCC/PVTT patients treated with radiotherapy. However, it does have some grammatical failures that should be corrected. Radiotherapy plays an increasingly important role in these patients and is included in the international guidelines as treatment option for Child Pugh A and B patients. SBRT appears to be even better than conventionally fractionated radiotherapy.

• Thank you so much for recognizing the importance of our work. We appreciate your efforts to enhance our work. Please accept an apology for any grammatical problems in the paper. We revised the manuscript with a stronger emphasis on the English language.

2. Patients who were treated between 2007 and 2019 were included. Radiotherapy has improved a lot in this period. Did you see any differences in local control, radiotherapy type and OS between patients treated earlier vs. later?

• We conducted more analysis focused on your advice and chose 2015 as a cut point. Because our advanced SBRT machines were installed that year. Table 1 and p11 lines 148-152 contained the data.

• 

3. Please correct EQD210 to EQD210

• We have fixed the error.

4. Please add the p value (site 10) to median survival comparing tumor size, also add the p value to survival comparing BED (site 11).

• As per your advice, we have included p-value.

5. Did you also know the ECOG-PS of these patients?

• We agreed that the important factor is ECOG-PS. ECOG-PS data were partially missing from medical records. We did our best to investigate the additional note and acquire the ECOG-PS data. Additionally, we do uni and multivariate analyses, which include the ECOG-PS. Tables 1 and 2 contain data.

6. Discussion:

The first paragraph is a copy of your introduction. Please remove in one of the 2 sections or adapt.

• The first paragraph has been removed from the discussion section. We recognized your issue and agreed that they shared a common meaning.

7. Tables 5, 6 and 7 are not uploaded. Please add.

• We regret for the errors. They have been corrected.

8. A part of your patients were treated with SBRT. Did you see a difference in LC and OS in this subgroup? Were these lesions smaller? Please add some more information on these SBRT patients in your results section.

• SBRT is becoming more widely employed and has a larger role in hepatocellular cancer. We conducted and summarized an analysis concentrating on SBRT patients. The data are included in the discussion section on page 19 line 291-296, as well as in the S3, S4 Table, and S1 Fig.

9. Please compare your data to the newly published SBRT data in the discussion section.

• We reviewed publications utilizing SBRT in HCC with PVTT. Our study demonstrated a comparable response rate and outcome. Page 19 line 297-300 and S5 Table provide the data.

REVIEWER 2 COMMENTS:

1. The manuscript analyses a large HCC cohort with portal vein tumor thrombus retrospectively. As data on this HCC subpopulation are relatively rare, the database is valuable, however the manuscript and data analysis have major issues which have to be fixed before publication. 

• We appreciate your time and effort in providing us with suggestions to improve the quality of our study. 

2. Radiation techniques and different dose-fractionation methods were completely mixed up and this can not be completely compensated by just calculating BED. Please add subgroup analyses of normofractionated and hypo fractionated (SBRT) treatments

• We conducted and presented an analysis focusing on SBRT patients and comparing them to the non-SBRT group, which includes hypo- and normofrationation patients. The data are given in the discussion section beginning on page 19 line 291-296, as well as in the S3, S4 Table, and S1 Fig.

3. What is meant by "advanced" RT technique? Techniques vary extremely from 3DCRT, IMRT, SBRT. Probably a subgroup analysis would bring more light to the data (eg group 1, SBRT; group 2, IMRT/VMAT/3DCRT with BED> a threshold and under this threshold

• We apologize for the confusion caused by the mix-up. What we are attempting to demonstrate is that the radiation approach used in the majority of this study, IMRT/VMAT, could spare liver tissue and can be safely delivered to HCC patients with PVTT who have a poor prognosis and limited treatment options. 

• After thoughtful consideration, we chose to do a subgroup analysis between SBRT and Non-SBRT, which includes normofractionation and hypofractionation utilizing 3D-CRT or IMRT/VMAT. However, the number of SBRT patients was insufficient to detect a significant difference in outcomes.

4. Statistical analysis: which software has been used? This info is missing from M&M

• They are now included on page 8 line 136.

5. Please provide local control and competing-risk adjusted local. recurrence rates.

• Competing risk analyses of local recurrence and associated factors have been conducted and are now presented on page 13 line 193-200 and in Table 3.

6. Toxicity part is very short. Please add non-hepatic toxicity (gastrointestinal, eg stomach, duodenum) which usually can happen by liver SBRT. How about central hepatobiliary toxicity, cholestasis, etc.

• We have included a section on gastrointestinal toxicity. However, our dataset did not include any grade 3-4 acute or late gastrointestinal adverse events. The cholestasis was not clearly identified. Although there were 10.3 percent grade 3 and 3.8 percent grade 4 increases in total bilirubin, the causes could have been liver decompensation, tumor development, or cholestasis. Table 6 contains the data.

7. Table 1: some numbers do not seem to be correct/ do not add on to 100%. there were 160 Patients, but some data are not provided for all of them eg CP-score only for 157, tumor size only for 159...).For TNM, even 165 M0 stages are reported (but only 160 patients).Please check, and provide information on missing data

• As a result of your feedback, we recalculated and included the missing data rows in Table 1.

8. Figure 2. probably the caption is mixed up. BED <10 cm?????

• We apologize for the error, which has now been corrected.

9. Figure 3. please add to the Kaplan Meier analysis a competing-risk adjusted local recurrence rate. probably we have. a survival time bias here with a lot of patients dying before local relapse.

• We present a competing-risk analysis of local relapse in Figure 5 and a comparison figure between patients who responded to radiation and those who did not, which was determined to be an unfavorable factor in Figure 6.

• Regretfully, due to limitations in statistical software, we were unable to include the number at risk beneath the figures.

10. Abstract: please explain "10"m do you mean alpha-beta ratio?

• It is the alpha/beta ratio used in calculation of BED. The further explanation is now included in the abstract. 

11. Do not begin a sentence with "and"

• We acknowledged and corrected all of them.

12. It is not usual to name the treating center in the abstract, please remove.

• We removed it on your recommendation.

13. Follow-up instead of follows-up.

• We have changed it in accordance with your recommendation.

14. What is meant by EQD210? probably EQD2- for alpha-beta 10?

• Yes, it is. It was our error not to subscript 10 after the EQD2. All of them have been fixed, and further explanations have been added to the materials and methods section.

Again, we appreciate you providing us with the opportunity to improve our manuscript through your insightful comments and queries. We have made a concerted effort to incorporate your suggestions. We hope that these adjustments will allow our manuscript to be published in PLOS ONE.

Yours Sincerely,

Chonlakiet Khorprasert M.D. (First author)

Division of Radiation Oncology, Department of Radiology, Faculty of Medicine, Chulalongkorn University, Pathumwan, Bangkok, Thailand.

Contact address: 1873, Rama 4 Rd., Pathumwan, Bangkok, Thailand 10330

Email: Chonlakiet.k@chula.ac.th

Napapat Amornwichet M.D., Ph.D. (Corresponding author)

Division of Radiation Oncology, Department of Radiology, Faculty of Medicine, Chulalongkorn University, Pathumwan, Bangkok, Thailand.

Contact address: 1873, Rama 4 Rd., Pathumwan, Bangkok, Thailand 10330

Email: Napapat.a@chula.ac.th, Napapat.amornwichet@gmail.com

---

## [Decision Letter · Decision Letter 1]

6 Sep 2021

Advanced radiotherapy technique in hepatocellular carcinoma with portal vein thrombosis: Feasibility and clinical outcomes

PONE-D-21-07678R1

Dear Dr. Amornwichet,

We’re pleased to inform you that your manuscript has been judged scientifically suitable for publication and will be formally accepted for publication once it meets all outstanding technical requirements.

Kind regards,

Panagiotis Balermpas

Academic Editor

PLOS ONE

Additional Editor Comments (optional):

Reviewers' comments:

Reviewer's Responses to Questions

**Comments to the Author**

1. If the authors have adequately addressed your comments raised in a previous round of review and you feel that this manuscript is now acceptable for publication, you may indicate that here to bypass the “Comments to the Author” section, enter your conflict of interest statement in the “Confidential to Editor” section, and submit your "Accept" recommendation.

Reviewer #1: All comments have been addressed

Reviewer #3: All comments have been addressed

2. Is the manuscript technically sound, and do the data support the conclusions?

Reviewer #1: Yes

Reviewer #3: Yes

3. Has the statistical analysis been performed appropriately and rigorously? 

Reviewer #1: Yes

Reviewer #3: Yes

4. Have the authors made all data underlying the findings in their manuscript fully available?

Reviewer #1: Yes

Reviewer #3: No

5. Is the manuscript presented in an intelligible fashion and written in standard English?

Reviewer #1: Yes

Reviewer #3: No

6. Review Comments to the Author

Reviewer #1: (No Response)

Reviewer #3: All reviewer comments have been addressed. Some English errors are still present ( sentences still begin with "and", eg page 12, line 150.

7. PLOS authors have the option to publish the peer review history of their article (what does this mean?). If published, this will include your full peer review and any attached files.

Reviewer #1: No

Reviewer #3: No

---

## [Editor Report · Acceptance letter]

14 Sep 2021

PONE-D-21-07678R1 

Advanced radiotherapy technique in hepatocellular carcinoma with portal vein thrombosis: Feasibility and clinical outcomes 

Dear Dr. Amornwichet:

I'm pleased to inform you that your manuscript has been deemed suitable for publication in PLOS ONE. Congratulations! Your manuscript is now with our production department. 

Kind regards, 

on behalf of

Dr. Panagiotis Balermpas 

Academic Editor

PLOS ONE